# Study on the CHZ-II Gravimeter and Its Calibrations along Forward and Reverse Overlapping Survey Lines

**DOI:** 10.3390/mi13122124

**Published:** 2022-11-30

**Authors:** Haibo Tu, Kun Liu, Heng Sun, Qian Cui, Yuan Yuan, Sunjun Liu, Jiangang He, Lintao Liu

**Affiliations:** 1State Key Laboratory of Geodesy and Earth’s Dynamics, Innovation Academy for Precision Measurement Science and Technology, Chinese Academy of Sciences, Wuhan 430071, China; 2School of Earth and Planetary Sciences, University of Chinese Academy of Sciences, Beijing 10004, China; 3School of Geospatial Engineering and Science, Sun Yat-sen University, Zhuhai 519082, China

**Keywords:** gravity measurement, calibration, forward and reverse overlapping lines calibration method, damping delay time, scale factor, measurement accuracy

## Abstract

The moving-base gravimeter is one of the key instruments used for Earth gravity survey. The accuracy of the survey data is closely related to the calibration precision of several key parameters, such as the damping delay time, the drift coefficient, the gravity scale factor, and the measurement accuracy. This paper will introduce the development of the CHZ-II gravimeter system in which a cylindrical sampling mass suspended vertically by a zero-length spring acts as a sensitive probe to measure specific force. Meanwhile, a GNSS (Global Navigation Satellite System) positioning system is employed to monitor the carrier motion and to remove the inertia acceleration. In order to achieve high-precision calibrations for the key parameters, a new calibration method performed along forward and reverse overlapping lines is proposed, which is used to calibrate the above parameters and to estimate the measurement accuracy of the instrument used for a normal gravity survey. The calibration principle and the shipboard calibration data processing method are introduced. The calibration was performed for three moving-base gravimeters and the corresponding results are determined, indicating that the method can significantly improve the accuracy of the parameters. For the CHZ-II gravimeter, the measurement accuracy of the survey is 0.471 mGal (1 mGal = 10^−5^ m/s^2^), which improved by 19.5% after applying the calibrated parameters. This method is also practical for use with aviation, marine and even vehicle-carried moving-base gravimeters.

## 1. Introduction

High-precision gravity data is of great significance for research in many fields [1,2,3]. The gravity datum and its change reflect the gravitational force received by a sampling mass at the measurement point, caused by the mass distribution, which contains valuable information. The gravity field can provide effective information for the study of land water storage, geological structures, and their evolutions. The gravity field in the ocean can be used to determine the ocean geoid, and to inversely calculate the geological structure of the seabed and the distribution of it resources. Ocean circulation information can also be deduced by combining this data with satellite radar altimetry data.

Gravity field data can be obtained by means of satellite gravimetry, marine/aviation gravity measurement, underwater gravity measurement, etc. The gravity satellite method is the most efficient among these approaches. It can measure the global change of the gravity field once a month, but can only determine the long-wave component (above 100 km) of the gravity field [4]. The gravity measurement approach carried out by ships or aircrafts (moving-base gravity measurement) strikes a good balance between measurement efficiency and inversion accuracy of the mass distribution, making it meaningful and indispensable [5,6].

A remarkable feature of the moving-base gravity measurement is that the carrier is highly maneuverable, and the interference of inertia acceleration can be millions of times larger than the abnormal gravity acceleration to be measured. In order to attenuate interference of the inertia acceleration, a damping system with an extremely large delay is usually adopted in moving-based gravimeters, and GNSS positioning technology is also used to record and determine the inertia acceleration of the carrier. There are two ways to suppress the disturbances from the carrier’s angular accelerations: applying the strap-down inertial navigation platform or the gyro-stabilized platform. Common accelerometers are used as a probe for gravimeters, including quartz accelerometers, cable-stayed spring accelerometers, atomic accelerometers, and vertical axis-symmetric accelerometers [7,8]. The technical solution adopted by the CHZ-II gravimeter described in this paper is a two-axis gyro platform employed to maintain the measurement attitude of the vertical axis-symmetrical probe, whose distinguishing feature is to suppress the strong inertial acceleration introduced by the carriers’ motion [9].

The calibrations of key parameters, such as delay time, gravity scale factor, time-related drift, and dynamic accuracy, for these moving-base gravimeters are very important for their application and high-precision data analyses. Many kinds of calibration methods have been developed to characterize the parameters of moving-base gravimeters, most of which are carried out in the laboratory state. Thus, the parameters obtained can be different from those needed for gravity survey conditions, since the working state of the gravimeter has been changed. Furthermore, some parameters can fluctuate with time. Therefore, calibration and inspection under the gravity survey state are necessary [10,11]. The baseline method, the tilt method, and/or the mass-load method are commonly used to calibrate their gravity scale factor before the gravimeters are used [12]; the tilt method and the change in Eötvös effect caused by the carrier’s heading direction and/or speed are usually used as fast inputs to test the delay time of the instruments [13]. However, these calibrations are not carried out in a normal measurement state. The dynamic measurement accuracy is estimated by the disagreement of many crossing points, or by the deviation of results along one-direction repeat data [10,14], but the damping delay time and scale factor cannot be calibrated using such methods. Reference [10] takes advantage of the change of the Eötvös effect caused by the east-west repeated survey lines as the input to calibrate the gravimeter scale factor, which improves the data processing accuracy, but it does not study the calibration of other parameters.

In the above context, this paper will introduce the principles and the progress of the CHZ-II gravimeter developed in our group. In order to improve the accuracy of the parameters of the instrument, a new calibration method, performed along forward and reverse overlapping survey lines, is proposed. The measurement also conforms to gravimetric specifications. The calibration method is addressed theoretically and is also used for experimental data processing, which shows that it can significantly improve the calibration accuracy of the damping time, facilitate the extraction of several parameters with high precision, and help to evaluate the instrumental accuracy in the normal measurement state. The method can be widely used in marine, aerial gravimetric, and vehicle-mounted gravimetric measurements before and after use. The calibration can be arranged in a straight line on the way to and back from the survey area, which is convenient and economical.

## 2. Development of CHZ-II Gravimeter

The CHZ-II is a kind of moving-base scale gravimeter, employing an axisymmetric damping structure as the sensitive probe, which is mounted on a two-axis gyro platform to maintain its measurement attitude. It inherits the advantages of the CHZ gravimeter [15]. As shown in Figure 1a, a cylindrical sampling mass is suspended by a main spring; two groups of horizontal strings are used to constrain the relative motion of the sampling mass with respect to the frame, except for the displacement along the axis of the main spring, which is sensitive to specific force. The mechanical structure and its testing have been previously reported [16]. The CHZ-II prototype has been constructed and systematically tested since 2012., The disturbance rejection characteristics due to the carrier motions have been analyzed [9].

In principle, the CHZ-II gravimeter works like a precise vertical electronic scale which can be suited to dynamic operating conditions. The probe employs a zero-length spring technology to suspend the cylindrical sampling mass on the frame. The zero-length spring technique is used to eliminate measurement errors caused by the self-weight of the spring. The tension of the spring balances the gravity of the sampling mass when the frame is in a static state. When the gravitational acceleration changes, the position of the sampling mass relative to the frame will be changed, which is monitored by a capacitive displacement sensor. A control signal from a proportional–integral–derivative (PID) controller is then sent to the electromagnetic feedback actuator so that the sampling mass is controlled back to its original position by the electromagnetic force. The acceleration variation can be expressed as $\Delta a=F_{e}/M$, which is the vector sum of inertial acceleration and gravitational acceleration in the opposite direction (i.e., specific force), where $F_{e}=BIl$ is the electromagnetic force on the sampling mass (mass *M*), B is the magnetic field strength in the gap of the permanent magnet, I is the feedback current, and l is the coil length of the actuator.

In order to ensure that the measurement error coupled from the horizontal acceleration is no more than 0.5 mGal, the alignment error of the two fixed points for all the wires in the height direction should be controlled within ±15μm, which raises a very high requirement for precision machining and assembly. One of the recent improvements is to integrate the main components of the gravimeter into a space of ϕ560 mm × 700 mm, as shown in Figure 1b. The control unit is used for communication among the gyro platform, the sensitive probe, and the GNSS positioning module, as well as data synchronization and storage.

## 3. Calibration along Forward and Reverse Overlapping Survey Lines

Enlightened by the idea of a multi-parameter joint extraction method for on-orbit gravity satellite systems, such as the GRACE mission [4], we propose a calibration method, performed along forward and reverse overlapping survey lines, to test and extract those key parameters of the gravimeter in the same state as a normal gravity survey.

The method collects data as follows: The moving-base gravimeter performs measurements along the same straight line twice, by forward and reverse sailing, respectively. The measurements conform to the gravity survey specification.

The data processing is analyzed as follows: in the forward measurement (called S1), considering the damping delay time Δt, the measured scalar gravity anomaly of any point on the survey line can be expressed by the following [10] (for vector gravimeters, each component can be expressed similarly):(1)$$g_{i}=g_{0}+K_{V}V_{t_{i}+\Delta t}-\delta E_{i}-\delta a_{v,i}-g_{0,i}-K_{d}\left(t_{i}-t_{0}\right),$$
where $g_{0}$ represents the gravity of the base point, $K_{V}$ represents the scale factor to convert the observation value to gravity acceleration, $t_{i}$ represents the moment that the gravimeter passes through the point i, $V_{t_{i}+\Delta t}$ is the output of the gravimeter at moment $t_{i}+\Delta t$; $\delta E_{i}$ and $\delta a_{v,i}$ are the Eötvös effect and vertical acceleration caused by the motion of the carrier at point i; $g_{0,i}$ represents the normal gravity, which is determined by the latitude of point *i*, $K_{d}$ represents time-related drift coefficient, and $t_{0}$ is the start time of the measurements.

Similarly, for the reverse measurement (called S2), the gravity anomaly measured through the same space point i for the second time can be expressed as follows (the corresponding quantities are represented by the symbols with ‘):(2)$${g_{i}}^{\prime }=g_{0}+K_{V}V_{{t_{i}}^{\prime }+\Delta t}-\delta {E_{i}}^{\prime }-\delta {a_{v,i}}^{\prime }-g_{0,i}-K_{d}\left({t_{i}}^{\prime }-t_{0}\right).$$

It should be noted that both $\delta E_{i}$ and $\delta {E_{i}}^{\prime }$ are given provided the GNSS positioning information of the carrier. In the aerial gravimetric measurement, an RTK positioning method differential with the base station is usually employed to achieve a correction accuracy of 0.5 mGal; however, in the marine gravimetric measurement, due to the slow movement of the carrier, the single-point precision positioning method can be satisfied, which can achieve the correction accuracy of 0.15 mGal, and there is no cumulative error in theory [8]; when there is an altitude difference between the two overlapping voyages in the forward and reverse measurements (because of the change of tide level during marine measurement, etc.), an error will be introduced to $\delta a_{v,i}$ and $\delta {a_{v,i}}^{\prime }$, considering the gradient effect of gravity. Therefore, it should be calculated to the same altitude to eliminate this error.

In this way, the two sequences of gravity data are matched according to the position. Since the gravity results at the same position must be the same, that is:(3)$$g_{i}={g_{i}}^{\prime },$$
then
(4)$$\left(\begin{array}{cc} V_{{t_{i}}^{\prime }+\Delta t}-V_{t_{i}+\Delta t} & t_{i}-{t_{i}}^{\prime } \end{array}\right)\left(\begin{array}{c} K_{V}\\ K_{d} \end{array}\right)=\delta {E_{i}}^{\prime }-\delta E_{i}.$$

Assuming that there are N pairs of points ($1\ldots i\ldots N,N>3,N\in N^{*}$) obtained in the measurement, one can acquire N equations as follows:(5)Bx=l,
where $B={\left(\begin{array}{cc} \vdots & \vdots \\ V_{{t_{i}}^{\prime }+\Delta t}-V_{t_{i}+\Delta t} & t_{i}-{t_{i}}^{\prime }\\ \vdots & \vdots \end{array}\right)}_{N\times 2}$, $x={\left(\begin{array}{c} K_{V}\\ K_{d} \end{array}\right)}_{2\times 1}$, $l={\left(\begin{array}{c} \vdots \\ \delta {E_{i}}^{\prime }-\delta E_{i}\\ \vdots \end{array}\right)}_{N\times 1}$.

In Formula (5), when Δt is uncertain, the adjustment cannot be calculated. Therefore, the calibration is proposed to be calculated in the following three steps, to obtain characteristic parameters Δt, $K_{V}$, and $K_{d}$, and then estimate the measurement accuracy.

Step 1. Calculate Δt. According to Equations (1) and (2), it is essential to search for an appropriate time $\Delta t_{x}\in \left(0,t_{\max}\right)$ for the gravity data relative to the GNSS positioning data within a certain time range, and shift the discrete gravity sampling sequences forward for both S1 and S2, according to the number of points:(6)$$m=\left[\Delta t_{x}\cdot f_{s}\right],$$
which gives the two new position-matched sequences g and $g^{\prime }$ the best correlation. In this way, the delay time of the gravimeter can be calculated by the number *m*. $f_{s}$ represents the data sampling rate. The normalized cross-correlation coefficient of the two sequences can be expressed as:(7)$$C_{\mathrm{fg}}[m]=\frac{\sum_{i=m+1}^{N-m}\left\{\left(g[i+m]-\bar{g}\right)\cdot \left(g^{\prime }[i-m]-{\bar{g}}^{\prime }\right)\right\}}{\sqrt{\sum_{i=m+1}^{N-m}{\left(g[i+m]-\bar{g}\right)}^{2}\cdot \sum_{i=m+1}^{N-m}{\left(g^{\prime }[i-m]-{\bar{g}}^{\prime }\right)}^{2}}},$$
where i+m and i−m means that the two sequences at the original position i are shifted forward by m points. The result of the above formula ranges from 0 to 1. Among all possible values of m limited by Formula (6), the *m*_0_ corresponding to the maximum of Formula (7) will be found, so that the best estimated value $\Delta \hat{t}$ is obtained, which is:(8)$$\begin{array}{c} \left[C_{\mathrm{fg}}[m_{0}],m_{0}\right]=\max\left(C_{\mathrm{fg}}[m]\right)\\ \Delta \hat{t}=m_{0}/f_{s} \end{array}.$$

It should be noted that the parameters, such as $K_{V}$ and $K_{d}$, have little impact on the calculation of Formulas (7) and (8), so the theoretical value can be adopted. Similar methods are also used to study the delay time between seismic signals.

Step 2. Calculate $K_{V}$ and $K_{d}$ by substituting $\Delta \hat{t}$ obtained from Step 1 into Formula (5), and then using indirect adjustment:(9)$$V=B\hat{x}-l,$$
one can get:(10)$$\hat{x}={\left(B^{T}PB\right)}^{-1}B^{T}Pl,$$
where $P={\left[\begin{array}{cccc} \sigma^{2}/\sigma_{1}^{2} & 0 & \cdots & 0\\ 0 & \sigma^{2}/\sigma_{2}^{2} & \cdots & 0\\ & & \vdots & \\ 0 & 0 & \cdots & \sigma^{2}/\sigma_{N}^{2} \end{array}\right]}_{N\times N}$ is the weight matrix, $\hat{x}={\left(\begin{array}{c} {\hat{K}}_{V}\\ {\hat{K}}_{d} \end{array}\right)}_{2\times 1}$ is the best estimate for the parameters to be measured; $\sigma^{2}$ and $\sigma_{i}^{2}$ are standard deviation for the unit weight and the statistical variance of the *i*-th point in the *l* sequence. The uncertainty of the estimates can be assessed by using the covariance matrix:(11)$$D\hat{x}={\left(B^{T}PB\right)}^{-1}\sigma^{2}=\left(\begin{array}{cc} {\hat{\sigma }}_{K_{V}}^{2} & \cdots \\ \cdots & {\hat{\sigma }}_{K_{d}}^{2} \end{array}\right).$$

Step 3. Substituting $\Delta \hat{t}$, ${\hat{K}}_{V}$, and ${\hat{K}}_{d}$ into Formulas (1) or (2), the observed gravity anomaly of the gravimeter at moment *t* is corrected as follows:(12)$${\hat{g}}_{t}=g_{0}+{\hat{K}}_{V}V_{t+\Delta \hat{t}}-\delta E_{t}-\delta a_{v,t}-g_{0,t}-{\hat{K}}_{d}\left(t-t_{0}\right).$$

The gravity anomaly discrepancy (also known as middle error) can be further calculated by Formula (13), which is used to evaluate the measurement accuracy and calibration quality of the gravimeter:(13)$${\hat{\sigma }}_{g}=\sqrt{\frac{\sum_{i=m_{0}+1}^{N-m_{0}}{\left({\hat{g}}_{i}-{{\hat{g}}_{i}}^{\prime }\right)}^{2},}{N-2m_{0}-1}}$$

## 4. Overview of Data Processing

According to the analysis in the previous section, the overview of real calibration data processing can be briefly described as follows: for time-series data S1 and S2 measured by the instrument, it is assumed that the GNSS altitude data has been used to correct $\delta a_{v,i}$, as well as the gravity gradient-related error, by applying prior parameters, such as scale factor and time-related drift coefficients. Firstly, taking the longitude and latitude of each sampling point of S1 as a reference, all points in the S1 and S2 series which fall into a certain range around the reference point (for example, the longitude and latitude change ±0.001°) are taken into account to calculate their averages as the gravity anomaly corresponding to the reference point. All points are retrieved by position in turn to form a new sequence g and $g^{\prime }$. The sequences of other data, including the time columns, are updated synchronously. Then, for each *m* value in Formula (6), Step 1 of the previous section is performed, and the correlation coefficient of the two gravity sequences is calculated according to Formula (7). Thus, according to Formula (8), the delay time $\Delta \hat{t}$ of the gravimeter is estimated. After using $\Delta \hat{t}$ to correct the original gravity data and performing Step 1 again, ${\hat{K}}_{V}$, ${\hat{K}}_{d}$, and ${\hat{\sigma }}_{g}$ can be obtained by following Step 2 and Step 3.

Figure 2 depicts the effect of the calibration process. The difference in specific force observations between the two voyages is theoretically equal to the difference introduced by the two Eötvös sequences. After deducting the Eötvös effect and inertial acceleration, the gravity values tend to be consistent, but due to the error of the damping delay and other coefficients, the two gravity sequences still cannot overlap. Since the gravity value of the same position should be the same, and which is not dependent on the measurement direction and when the measurement is taken, the parameter error can be fitted and further corrected accordingly. Finally, the gravity value of the two voyages should coincide within the measurement error range, as shown by the black dotted line in the figure.

## 5. The Process of Data Calibration

A standard calibration procedure for the CHZ-II gravimeter has been carried out by Ship XYH06 using the above-mentioned method [17]. The straight survey line changes by 1° in longitude and 2.5° in latitude, and has a length of about 300 km. The experiment has a sailing speed of 9.5–11 kn, a data sampling rate of 1 Hz, a one-way measurement lasting about 17 h, and a total measurement duration of 35 h. Those data points that cannot find a position-matching point within the threshold of ±0.001° in latitude and longitude (around 100 m horizontally) are discarded so that the part with a poor correspondence to the survey lines has no effect on the calibration result. There are two other moving-base gravimeters for measurement, GT-2M and LCR-S, in the same cabin on the ship; the same processing method is also used for data analysis for comparison.

Figure 3 shows the output results of the three instruments on the S1 (forward) and S2 (reverse) survey lines. The empirical value of the scale factor is used in the calculation of the gravity anomaly. CHZ-II had not corrected the damping delay time yet, while the other two instruments had corrected the delay according to the empirical value. For the convenience of observation and comparison, the results of different instruments are separated by 10 mGal along the vertical axis. Taking CHZ-II as an example, it can be seen that before the parameter corrections, the gravity data contains an observable delay lag and a scale factor error.

Figure 4 shows the correlation coefficients of the S1 and S2 survey lines calculated by shifting the gravity measurement data forward and backward relative to the GNSS data, along with its impact on the deviation of S1–S2. As shown on the left panel, when the correlation coefficients become the largest, the gravity data best match the position sequences. The maximum correlation coefficients for the forward and reverse gravity data of the three instruments are 0.996, 0.999, and 0.962, respectively. Then the delay times are calibrated as $\Delta \hat{t}=m_{0}/f_{s}$, with *m*_0_ corresponding to the shifted point that best matches the GNSS position for those instruments, respectively. It can also be seen from the right panel that, when the deviation between S1 and S2 sequences is the smallest, it also implies that the data match the best. The minimum deviations are 0.585 mGal, 0.290 mGal, and 1.729 mGal, respectively. The required delay correction time is consistent with the result obtained from the right panel.

To extract the other parameters, substitute the extracted damping delay time into the original gravity measurement data, and process the following fitting. By using the calibration results to recorrect the original data, one can obtain the final results of the three instruments on the S1 and S2 survey lines, as shown in Figure 5.

The calibration results are listed in Table 1. It can be seen that the calibration accuracy $\Delta \hat{t}$ is sufficient for the current moving-base gravity measurement; the closer the scale factor is to 1, the more accurate the empirical coefficient of the instrument. The maximum relative error before calibration can reach 1.7%. In the current calibration, the change of the Eötvös effect between S1 and S2 is 60 mGal, and the error is 0.15 mGal. Therefore, the relative calibration accuracy of the scale factor is limited to 0.3%, although the fitting error is as small as 2 × 10^−4^. If the calibration survey line is designed along the east-west direction and the speed reaches 20 kn, the relative accuracy of the scale factor can be improved to 0.05%. The CHZ-II gravimeter has a drift coefficient of 0.173 mGal/day, which shows that it is essential to extract the drift coefficient in the gravity survey state and then correct it during the data processing. The measurement accuracy of the instruments is evaluated after using the calibrated parameters, as shown in the ${\hat{\sigma }}_{g}$ column. Compared with the deviation values (extracted from Figure 4) before the application of the calibrated parameters, it can be seen that the above calibration method can improve the measurement accuracy of the instruments. Among these, the drift of CHZ-II has a significant impact on the application of the original data, the white noise of GT-2M is small, and the measurement accuracy after calibration can reach 0.471 mGal and 0.199 mGal, respectively. The above calibration improves the measurement accuracy of the three instruments by 19.5%, 31.4%, and 7.8%, respectively.

## 6. Summary and Discussion

This paper introduces the development of the CHZ-II moving-base gravimeter, which is constructed with a cylindrical restrained sampling mass as the sensitive probe, mounted on a two-axis gyro platform to maintain its measurement attitude. Its working principle and mechanical assembly requirement are addressed. In order to improve the application accuracy of the gravimeter, a new calibration method is proposed which conforms to the normal gravity survey specification. The calibration is performed along forward and reverse overlapping survey lines. With the data processing method described above, several key parameters of the instrument can be more accurately evaluated. The method is quite suitable to accurately extract the damping delay time. Different data sections and different point-matching thresholds always result in the same delay time. The calibration results from three instruments show that the method can improve the application accuracy of the instruments. The calibration method can also be applied to marine and aviation surveys using other gravity measurement systems.

The existing calibration methods usually have limited sampling points to fit the key parameters. As a comparison, this method uses a large number of matching points distributed on the same track during the gravity survey and can also merge matching-points from different data sections to extract those parameters simultaneously. Thus, the matched data contain more instrumental information and result in more accurate parameters. Meanwhile, it helps to shorten the calibration duration and reduce the cost.

The effect of the above calibration method is essentially similar to that of the traditional gravimetric network adjustment. Nevertheless, the calibration process directly obtains more accurate parameters, in addition to the estimate of gravity measurement accuracy. In addition, several parameters can be extracted simultaneously by the same calibration data obtained in the course of the gravity survey. Therefore, this method has a wide application. However, it should be noted that this method only realizes the calibration of the scale factor under a certain gravitational difference within the scope of the survey area. Since the instrument may have non-negligible nonlinearity, for a larger area or even the global gravity measurement (the maximum gravity change can reach 5000 mGal), the calibration scale factor still depends on the establishment of an accurate full-scale nonlinear model, or alternatively, the above calibration voyages must be performed again.

As to the planning of an actual calibration survey line, some aspects can be further considered: (1) an obvious gravity anomaly along the survey line is helpful for calibrating the delay time of the instrument; (2) a larger change in specific force caused by the Eötvös effect is helpful to lower the calibration error of the scale factor; (3) good survey conditions may be selected in order to extract the calibration parameters with high precision, but from the perspective of evaluating the measurement accuracy of the instrument in practice, the above-mentioned measurement process and accuracy evaluation under general, and even severe, working conditions are also of practical significance.

## Figures and Tables

**Figure 1 micromachines-13-02124-f001:**
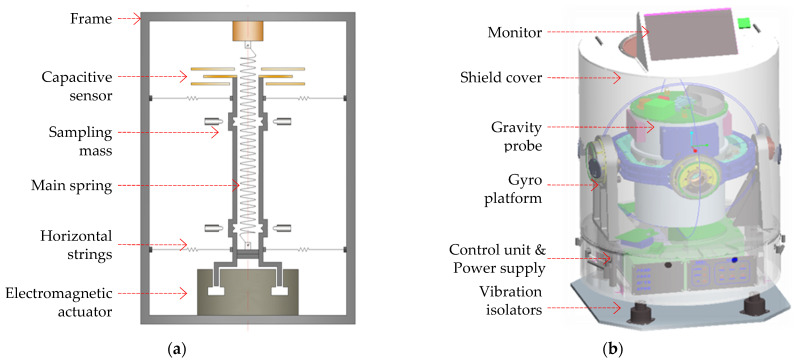
CHZ-II gravimeter. (**a**) Schematic diagram of the gravity probe, which is constructed with a cylindrical sampling mass suspended by a spring. Two groups of horizontal strings are used to constrain the motion of the sampling mass with respect to the frame, except for the displacement along the axis of the spring. An electromagnetic actuator is used for the feedback control of the sampling mass. (**b**) System components, with the probe supported by a two-axis gyro platform.

**Figure 2 micromachines-13-02124-f002:**
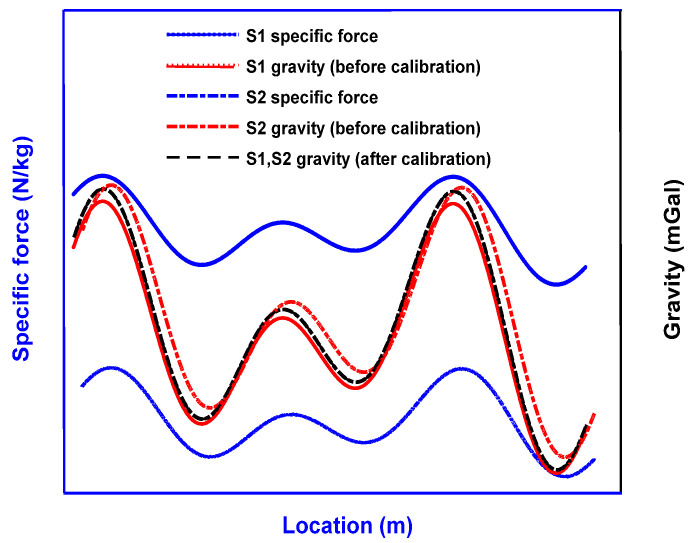
Schematic diagram illustrating the calibration principle. Successful calibration will bring both S1 and S2 sequences to the same real gravity anomaly curve, as shown in the black dashed line.

**Figure 3 micromachines-13-02124-f003:**
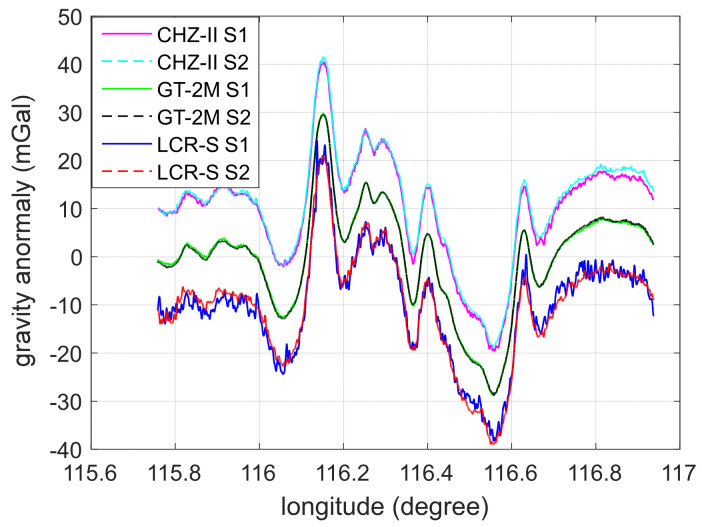
The gravity anomaly measurement results of the three sets of instruments on the S1 and S2 survey lines after matching them by position.

**Figure 4 micromachines-13-02124-f004:**
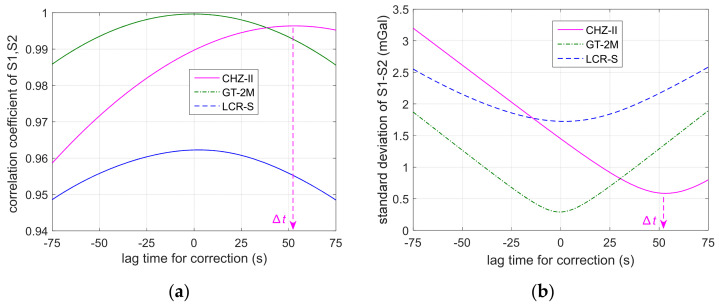
The influence of the damping delay time on the S1 and S2 sequence matching: (**a**) correlation coefficients; (**b**) deviation between the two measurement residuals.

**Figure 5 micromachines-13-02124-f005:**
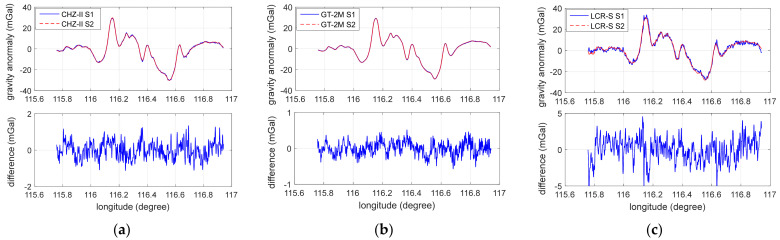
The gravity values and the mutual differences of the three instruments on the S1 and S2 survey lines after applying the calibrated parameters: (**a**) CHZ-II; (**b**) GT-2M; (**c**) LCR-S.

**Table 1 micromachines-13-02124-t001:** Calibration results for three types of moving-base gravimeters.

Model	Damping Delay Time $\Delta \hat{t}$ (s)	Scale Factor ${\hat{K}}_{V}$	Drift ${\hat{K}}_{d}$ (mGal/d)	Measurement Accuracy ${\hat{\sigma }}_{g}$ (mGal)
CHZ-II	53 ± 1	1.002 ± 0.003	0.173 ± 0.009	0.471
GT-2M	−1 ± 1	1.011 ± 0.003	0.085 ± 0.009	0.199
LCR-S	2 ± 1	1.017 ± 0.003	−0.013 ± 0.007	1.595

## Data Availability

The datasets analyzed during the current study are available from the corresponding authors on reasonable request.

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
