# Peer review of "Study on the CHZ-II Gravimeter and Its Calibrations along Forward and Reverse Overlapping Survey Lines"

_micromachines, 2022, doi:10.3390/mi13122124_

Round 1
Reviewer 1 Report
This article addresses a newly-developed moving-base gravimeter–CHZ-II and the related gravity surveys with repeated lines have been carried out so that a new calibration method for some key parameters has been presented. Firstly, the zero-length spring has been adopted to implement the sensitive gravity probe. Combined with the two-axis gyro-platform, the dynamic measurement system of gravity is integrated. The actual marine gravity surveys have demonstrated that the system could be well performed. In the term of calibration method, precise determination of key parameters in real dynamic environment is of great importance to improve the performance of moving-base gravimeters. The authors introduce a new calibration approach and the corresponding data-processing procedures, which allow the extraction of few key parameters of the gravimeter with the same set of data. Besides, the practical surveys of gravity based on the new integrated gravimeters were carried out and the experimental results were analyzed, which show that the new developed method of calibration could improve the accuracy of some key parameters significantly so that the performance of the gravimeter has been checked. This method seems more efficient and may be helpful for the improvement of the performance of moving-base gravimeters. Hence, the referee recommends this paper is accepted for publication.
Here are some comments that the author could address.
1. Abstract: The detailed performance of CHZ-II should be given, especially after the calibration, such as the accuracy of key parameters, the measurement accuracy of the surveys.
2. Keywords: CHZ-II Gravimeter is not the keyword, and should be removed.
3. Introduction, Line 60: The comparison with other calibration methods should be given in details.
4. Section 2, Figure 1: The horizontal strings could be introduced, and the relationship with the two groups of wire should be explained.
5. Line 108: “PID” should be the full name.
6. Formula (1): The terms should be explained in order and more detailed.
7. Figure 2: This figure could be improved. The units should be given and the capture should not cover the curves. Moreover, the sentence of “Figure 2 depicts the effect of the calibration process.” is confusing.
8. Table 1: The damping delay time and drifts of the CHZ-II gravimeter seems much larger than the other two commercial marine gravimeters, the explanation could be introduced.
9. Is this method of calibration suitable for cross-lines marine survey?
10. Overall, the words of “the” should be checked since they are misused in many places.

Reviewer 2 Report
Comments on “Study on CHZ-II Gravimeter and Its Calibrations along Forward and Reverse Overlapping Survey Lines (No.1973938)” by Tu et. al. This study describes the development of the CHZ-II moving-base gravimeter, which has two main characteristics: First, a cylindrical sampling mass suspended vertically by a zero-length spring acts as a sensitive probe to measure specific force; Second, a GNSS positioning system is employed to monitor the carrier motion and to be able to remove the inertia acceleration. Furthermore, to improve the application accuracy of the CHZ-II gravimeter, this study proposes a new calibration method performed along forward and reverse overlapping lines, which can calibrate the key parameters (i.e., damping delay time, drift coefficient, gravity scale factor) and estimate the measurement accuracy in the normal gravity survey specification. The calibration experiment was performed for three moving-base gravimeters, the results indicate that the calibration method can significantly improve the accuracy of those key parameters. I believe that the goal of this study is worth pursuing, and the described experiment and method can be informative to the readers of Micromachines. So, I suggest that this work could be published after a minor revision.
The major comments: In addition to the comparison of measurement results, I think the authors should supplement the brief introduction of GT-2M and LCR-S gravity measurement instruments.
The detailed comments:
1. Lines 13 and 18: I think these “several key parameters” should be presented when they are mentioned for the first time.
2. Lines 38 and 39: “gravity satellite” should be “satellite gravimetry”.
3. Line 86: “help” shold be “help to”.
4. Line 108: abbreviation of “PID”, please give the full name.
5. Line 120: “φ560mm × 700mm”, please correct it.
6. Lines 148 and 150: “0.5 mGal” and “0.15 mGal”, please explain how these accuracies are obtained, or provide some references for it.
7. Line 160: the subscript should be “N×2” and “N×1”.
8. Line 202: “step 1)” should be “step 1”.
9. Line 207: for the Figure 2, the legend covers the curves, please redraw it.
10. Line 248: “from right that” should be “from the right panel that”.
Reviewer 3 Report
Please see my full report attached.
